# Effects of Diabetes Mellitus-Related Dysglycemia on the Functions of Blood–Brain Barrier and the Risk of Dementia

**DOI:** 10.3390/ijms241210069

**Published:** 2023-06-13

**Authors:** Mateusz Wątroba, Anna D. Grabowska, Dariusz Szukiewicz

**Affiliations:** Laboratory of the Blood–Brain Barrier, Department of Biophysics, Physiology & Pathophysiology, Medical University of Warsaw, Chałubinskiego 5, 02-004 Warsaw, Poland; mateusz.watroba@wum.edu.pl (M.W.); anna.sepulveda@wum.edu.pl (A.D.G.)

**Keywords:** diabetes mellitus, blood–brain barrier disruption, dysglycemia, hyperglycemia, insulin resistance, neurodegenerative disease, diabetic complications, dementia, diabetic encephalopathy

## Abstract

Diabetes mellitus is one of the most common metabolic diseases worldwide, and its long-term complications include neuropathy, referring both to the peripheral and to the central nervous system. Detrimental effects of dysglycemia, especially hyperglycemia, on the structure and function of the blood–brain barrier (BBB), seem to be a significant backgrounds of diabetic neuropathy pertaining to the central nervous system (CNS). Effects of hyperglycemia, including excessive glucose influx to insulin-independent cells, may induce oxidative stress and secondary innate immunity dependent inflammatory response, which can damage cells within the CNS, thus promoting neurodegeneration and dementia. Advanced glycation end products (AGE) may exert similar, pro-inflammatory effects through activating receptors for advanced glycation end products (RAGE), as well as some pattern-recognition receptors (PRR). Moreover, long-term hyperglycemia can promote brain insulin resistance, which may in turn promote Aβ aggregate accumulation and tau hyperphosphorylation. This review is focused on a detailed analysis of the effects mentioned above towards the CNS, with special regard to mechanisms taking part in the pathogenesis of central long-term complications of diabetes mellitus initiated by the loss of BBB integrity.

## 1. Diabetes Mellitus—A Short Resume

Diabetes, also known as diabetes mellitus (DM), is a group of the most common endocrine/metabolic diseases worldwide, characterized by sustained high blood glucose levels [1]. It results from a deficiency of insulin actions, either because of insulin deficiency or because of insulin resistance [2]. It impairs, mainly, glucose metabolism, but it results in a disease affecting almost each tissue in the organism. Complications of diabetes mellitus may include myocardial infarction, blindness, chronic renal failure, neuropathy, and loss of cognitive functions [3]. Impaired transport of glucose to insulin-dependent cells results in glucose excess in extracellular fluids and in insulin-independent cells, which is accompanied by severe deficiency of glucose within insulin-dependent cells. Thus, some manifestations of diabetes mellitus are related to glucose depletion inside insulin-dependent cells, others are related to glucose excess toxicity in relation to other cells and tissues, and more—to oxidative stress or using alternative substances as energy sources. Many long-term complications of diabetes mellitus depend on disorders of blood vessels, which can be divided into those referring to large vessels (macroangiopathies) and those referring to small vessels (microangiopathies). Diabetic macroangiopathies increase the risk of atherosclerosis and its related cardiovascular diseases, while diabetic microangiopathies can lead to chronic renal failure, neuropathy, and blindness, among others. While diabetic retinopathy is regarded as the most common cause of blindness in western countries, effects of diabetes mellitus towards the blood–brain barrier (BBB) are much less studied. It is quite astonishing, as retina is a specialized part of the central nervous system (CNS), and thus the blood-retina barrier is regarded as a specialized region of the blood–brain barrier [4]. The optic nerve is a part of CNS; thus, its capillaries carry out some barrier functions. Complications of DM in relation to BBB share many common features with those referring to diabetic retinopathy.

There are two main kinds of diabetes mellitus—type 1 DM and type 2 DM. Type 1 DM is usually an acute onset disease of young people, underlied by the autoimmune destruction of beta cells of pancreatic islets (islets of Langerhans) and a subsequent deficiency of insulin. Type 2 DM is usually a chronic disease with an insidious onset, underlied by tissue insulin resistance and a resulting deficiency of insulin actions, despite a quite decent concentration of insulin in the plasma. Type 2 DM is strongly associated with obesity and accounts for 90% of DM cases worldwide. The incidence of type 2 DM has been increasing for about 40 years and nowadays affects about 9% of people worldwide [5]. This refers especially to developed countries and elderly people because type 2 DM is strongly correlated with a high-sugar diet, which leads to gradual complications, even when developing; therefore, its prevalence increases with age [1].

## 2. Blood–Brain Barrier

The blood–brain barrier has been described in detail, both morphologically and physiologically [6]. At the cellular level BBB is formed by cerebral capillary endothelial cells and the closely apposed astrocyte end-feet processes [7]. Endothelium within BBB shows a specific pattern of expression of transmembrane transport systems, regulating the transport of various substances inside and outside the cerebrospinal fluid [8,9]. BBB endothelial cells are connected to one another with tight junctions, are devoid of fenestrations, and hardly perform pinocytic transport, which promotes intracerebral environment homeostasis and is a unique trait of endothelium within cerebral microcirculation. Tight junctions connecting the adjacent endothelial cells comprise a diffusion barrier that prevents most plasma components, such as electrolytes, ionized xenobiotics, and other water-soluble substances, from crossing BBB through a paracellular route. On the other hand, the co-existing specialized efflux transport mechanisms, dependent on such proteins as P-glycoprotein (P-gp), breast cancer resistance protein (BRCP), and multidrug resistance protein (MRP-4), regulate the trafficking of amphipathic and hydrophobic substances across BBB to protect the brain from the exposal of potentially toxic substances [10]. The BBB is quite important in the context of diabetes because hyperglycemia can disrupt its structure and function through inducing oxidative stress and secondary inflammatory response within its components, while hypoglycemia may alter the rate of glucose and the transportation of amino acids across the BBB, thus affecting brain functioning and biosynthesis of some neurotransmitters [9,10].

### 2.1. Glucose Transport across BBB

Glucose transport across the BBB may depend on two independent classes of glucose transporters: facilitated diffusion transporter proteins (GLUTs) and secondary active transporter proteins, also called sodium-glucose cotransporters (SGLTs) [11,12]. GLUTs include 14 proteins, the first identified of which, and the one mainly accounting for glucose transport across the BBB, is GLUT-1 [13,14,15]. GLUT proteins are saturable glucose carriers, transporting glucose down its concentration gradient [16,17]. While GLUT-1 molecules present in astrocytes and endothelial cells transport glucose across BBB, GLUT-3 is thought to be the most important glucose carrier transporting glucose from the cerebrospinal fluid to the neurons, although not the only one [14]. In mature and completely differentiated endothelial cells, GLUT-1 transporters are not homogenously distributed, and more of them can be found at the luminal part of their cell membrane (i.e., adjacent to the lumen of brain capillaries) than at the opposite (abluminal) part [18,19].

Increased glucose transport across BBB is correlated with increased luminal expression of GLUT-1 molecules. On the other hand, increased abluminal expression of GLUT-1 molecules is observed when there is a downregulation of GLUT-1; in such a situation, the luminal expression of GLUT-1 molecules is reduced, while its abluminal expression rises [20], suggesting that an altered distribution of GLUT-1 molecules can modulate glucose transport from the plasma to the cerebrospinal fluid.

GLUT-4, an insulin dependent glucose transporter, together with insulin receptors located within the endothelial cells of brain capillaries, can also play a significant role in glucose transporting to the CNS. In normal conditions, insulin is hardly synthesized in the brain, but insulin transport across the BBB has been found to increase in the course of DM [21,22,23,24]. Insulin can get to the brain through the cerebrospinal fluid, reaching it within locations devoid of the BBB, e.g., in circumventricular organ, or crossing the BBB via insulin receptors also acting as insulin transporters [25,26]. A schematic diagram of the BBB, with participation of individual cells in glucose transport, is presented in Figure 1.

Effects of insulin action towards the brain are interesting in the context of discovery, suggesting that DM can promote Alzheimer’s disease (AD) due to mechanisms including the attenuation of intracellular-insulin receptor-dependent signaling within the BBB. Although complete elucidation of those mechanisms may require some additional studies, insulin may play a role in regulating tau protein phosphorylation in the course of tauopathies, through glycogen synthase kinase-3 (GSK-3) activation and stimulating the expression of soluble receptors for advanced glycation end products (s-RAGE) [29,30]. A high expression of these receptors is correlated with an increased incidence of ischemic heart disease as a complication of DM, as well as with higher mortality due to all complications of DM, both type 1 and type 2 [31,32]. Higher levels of these receptors can also reflect the expression of tissue receptors for advanced glycation end products (RAGE) in the course of DM [33]. The advanced glycation end products (AGE)-RAGE system activation seems to play a crucial role in the pathogenesis of DM-related angiopathy and thrombosis [34,35]. Disruption of BBB integrity and its function may have a profound effect on the CNS, thus being a prodromal manifestation of serious neurological disorders. DM-related disruption of BBB integrity and its function can have an essential effect on the CNS.

#### 2.1.1. Hyperglycemia and Its Effect on Glucose Transport across the BBB

On the basis of general physiology, it is thought that when a substance occurs in excess, expression of receptors for the substance falls, undergoing an effect known as downregulation. These effects are aimed at homeostasis maintenance, in terms of substrates delivery to the cells, and adjusting it to the cell demand. Therefore, it is suggested that the biological response to hyperglycemia may include the downregulation of proteins accounting for glucose transport from plasma to peripheral tissues. The downregulation of glucose transporters, in response to hyperglycemia, has been confirmed in some research studies [36,37,38] but not in others [39,40,41,42,43].

Studies on the expression of glucose transporters in rats with streptozotocin-induced DM show that chronic hyperglycemia reduces both GLUT-1 and GLUT-3 expression at the level of transcription and translation; the downregulation of these transporters in vivo is observed regardless of the method of DM induction [37]. Local brain-glucose utilization rises in the course of chronic, but not acute, hyperglycemia in Sprague Dawley rats. It is accompanied by a moderate, yet clinically significant, fall of GLUT-1 expression in cerebral capillaries, but not of GLUT-3 expression [44]. Regardless of its effect on glucose transport, hyperglycemia can disrupt both structure and function of the BBB through promoting oxidative stress and secondary inflammation [5,10].

#### 2.1.2. Hypoglycemia and Its Effect on Glucose Transport across the BBB

Hypoglycemia can be a major threat in the course of DM, since it can result in a serious disruption of CNS functions. Frequent attempts of reducing the plasma glucose concentration in patients with DM may lead to hypoglycemia and hypoglycemic damage of the brain cells, which can induce hypoglycemia-associated autonomic failure (HAAF). General concept of HAAF background assumes that adrenalin release from adrenal medulla, which normally would inhibit insulin secretion and boost glucagon secretion, is blunted in the course of DM. This can be accompanied by being unaware of hypoglycemia, which can lead to a vicious circle of recurrent hypoglycemia and further impairment of hypoglycemia-preventing mechanisms. Although the clinical significance of HAAF is widely known, its detailed mechanisms and mediators remain grossly unknown [45,46]. Despite these gaps in our knowledge, hypoglycemia and HAAF can be avoided both by using continuous blood–glucose monitoring devices and by applying modern medications in type 2 diabetes, which are less likely to cause rebound hypoglycemia than older ones [46]. Most DM patients have difficulties with the maintenance of normal plasma glucose concentration because of HAAF and hypoglycemia unawareness. Diabetic rat exposal to acute hypoglycemia does not lead to an increased glucose concentration in the extracellular fluid of the inferior colliculus [47], thus showing that short-lasting, recurrent hypoglycemia will affect neither glucose transport to the brain or brain–glucose metabolism rate. However, an increased expression of GLUT-1 within the BBB, at the level of transcription and translation, in response to chronic hypoglycemia, indicates the existence of a compensatory mechanism, increasing glucose transport across the BBB in case of chronic hypoglycemia [48].

Increased glucose transport across the BBB in response to chronic hypoglycemia may result both from an increased expression of GLUT-1 and the redistribution of GLUT-1 molecules within the BBB [48]. The brain–glucose uptake in vivo, in response to chronic hypoglycemia, increases regardless of the hypoglycemia-inducing method [49]. Similar observations (glucose concentration in the brain increased by 48%) can be made in rats exposed to hypoglycemia for 12–14 days [49]. Furthermore, hypoglycemia results in the increase of brain–glucose uptake by 25–45%, as well as an increased GLUT-1 expression by 23% and the redistribution of GLUT-1 molecules from the abluminal to luminal part of endothelial cells within the BBB [39].

Increased GLUT-1 expression, combined with the redistribution of its molecules to the luminal pole of endothelial cells, can additionally aggravate hypoglycemia through stimulating glucose transport across the BBB. In addition, acute or mild hypoglycemia stimulates the expression of GLUT-1, GLUT-4, angiotensinogen- and mitogen-activated protein kinase (MAPK) phosphatase-1 [50]. An increased expression of angiotensinogen may promote vasodilation with locally increased blood flow. This may, in turn, locally elevate glucose concentration, resulting in an overestimation of the blood–glucose level by the hypothalamus and, subsequently, inhibit hypothalamic mechanisms aimed at counteracting hypoglycemia. However, falls in plasma glucose concentration, which occurs repeatedly in a long-term perspective, may promote deficits in attention span and increase the risk of depression [51].

### 2.2. Alterations in Amino Acid Transport across the BBB in the Course of DM

Choline is the precursor of acetylcholine—a neurotransmitter taking part in muscle control and memory, among others. Choline is transported to the brain by a saturable transporter acting within the BBB. Prolonged hyperglycemia can inhibit choline transport across the BBB [52]. As to other amino acids, the transport of branched chain-neutral amino acids is increased in DM, while the opposite is true for basic amino acids and some essential amino acids, such as methionine, lysine, phenylalanine, and tryptophane. Interestingly, these alterations in amino acid transport across the BBB result from their altered concentration in the plasma, not from the altered function of amino acid transporters across the BBB [53].

### 2.3. Effects of DM on BBB Integrity and Permeability

Both in vitro and in vivo studies show that DM disrupts BBB integrity, which results in its increased permeability [54,55,56,57]. In vitro studies, using the co-culture of human brain microvascular endothelial cells and juxtaposed astrocytes, indicate a loss of BBB integrity (evaluated on the basis of transendothelial electric resistance) when hyperglycemia-mimicking conditions (glucose concentration in the medium elevated to 25 mmol/L) are maintained for at least 5 days. The integrity can be brought back to a normal level after the normoglycemia-mimicking conditions are restored, or when antioxidants are added [58]. In other studies [59], hyperglycemia has been found to significantly increase the expression of pro-inflammatory cytokines (TNF-α, IL-1, IL-4, IL-6), which is followed by the upregulation of NF-κB and the signal transducer and activator of transcription 3 (STAT-3) proteins. There are also research studies performed on brain slices obtained from diabetic rats, showing an impaired communication between astrocytes and endothelial cells through gap junctions and an increased production of reactive oxygen and nitrogen species, although in an unclear mechanism [60,61]. In addition, an increased production of VEGF has been observed in response to the presence of advanced glycation end products [62].

A high glucose concentration (30 mmol/L) in endothelial cell cultures stimulates the expression of hypoxia-inducible factor 1 alpha (HIF-1α) transcription factor and vascular endothelial growth factor (VEGF), as its downstream effector. VEGF enhances GLUT-1 molecule translocation to the cell surface within the BBB, as well as downregulates proteins that are responsible for the proper functioning of tight junctions between endothelial cells (e.g., zonula occludens-1 (ZO-1) and occludin) [63]. In this way, hyperglycemia may increase BBB permeability. Occludin expression in brain microcapillaries in diabetic mice is significantly lower than in their healthy counterparts [64]. The inhibition of VEGF expression normalizes the expression of occludin and ZO-1, thus attenuating interendothelial leakage. A similar effect can be achieved through the inhibition of HIF-1α activity, which normalizes BBB integrity and permeability [65].

In humans, research study results are a bit different. The human plasma VEGF level falls in hyperglycemia and rises in hypoglycemia, which may constitute a neuroprotective mechanism, maintaining a constant rate of glucose influx to the brain [66]. Such findings implicate a key role of VEGF as a regulator of vascular permeability in the course of BBB dysfunction due to hypo or hyperglycemia. Regardless of VEGF, recent studies suggest a significant role of matrix metalloproteinases (MMPs) in the course of dysglycemia-induced loss of BBB integrity. In diabetic rats, BBB permeability for 14C radiolabeled sucrose rises in response to increased MMP-2 expression, which is accompanied by a reduced expression of occludin and ZO-1 [54,67,68,69]. Advanced glycation end products, similarly to VEGF, stimulate MMP-2 release [62].

Acute transient hyperglycemia may also induce inflammatory response and endothelial damage, which has been studied on rat models of ischemia-reperfusion damage. An increased expression of HMGB1 and intercellular adhesion molecule 1 (ICAM-1) has been found as a result of ischemia-reperfusion damage, both with co-existing mild hyperglycemia (plasma glucose concentration = 150 mg/dL) and with transient severe hyperglycemia (plasma glucose concentration = 400 mg/dL). These effects have been correlated with a disrupted integrity of the BBB [70]. In addition, an increased expression of ICAM-1 is accompanied by an increased expression of IL-1β in diabetic rats after 3 days of reperfusion [69].

A hyperglycemia-induced expression of HIF-1α and VEGF act synergistically, enhancing the detrimental response of the BBB to occlusion and stopped blood flow [71], thus additionally contributing to the loss of BBB integrity observed during reperfusion, although this hypothesis requires confirmation. Both brain capillary permeability and the level of pro-inflammatory cytokines rise in DM type 1 patients with ketoacidosis. Diabetic ketoacidosis, especially pronounced in the course of type 1 DM, increases BBB permeability and promotes complications, such as brain edema [72]. What can be observed in a similar study is the absence of proteins responsible for the proper functioning of tight junctions, such as occludin, claudin-5, ZO-1, and junctional adhesion molecule-1 (JAM-1), as well as albumin extravasation and an increased expression of inflammatory response markers, such as NF-κB, C-C motif chemokine ligand 2 (CCL-2), and nitrotyrosine. This indicates that neuroinflammation combined with the loss of BBB integrity plays an essential role in the pathogenesis of brain edema in the course of diabetic ketoacidosis [68]. All these findings suggest that DM significantly impairs BBB integrity and the maintenance of CNS homeostasis, which may promote the risk of major neurological disorders.

Hyperglycemia can also impair vitamin C delivery, both to the retina and to the brain. Vitamin C is transported across the BBB with a GLUT-1 transporter, in the form of dehydroascorbic acid (DHA), subsequently transformed to ascorbic acid. Vitamin C is required for the biosynthesis of collagen, catecholamines and peptide neurotransmitters. In rats, with streptozotocin-induced DM, the transport of DHA to the brain is reduced by 84.1% [10]. The expression of P-glycoprotein (a glycoprotein ATP-binding cassette transporter) may also be altered in diabetic mice [73,74,75] and brought back to normal with insulin treatment [74].

Additional histologic alterations and vascular abnormalities, observed within cerebral microcirculation in the course of DM, may include the thickening of the capillary-basal membrane, collagen deposition, accumulation of lipid-peroxidation byproducts, and endothelial degeneration [76]. These alterations may promote aberrant neovascularization and brain remodeling, which can contribute to vascular damage, an increased risk of hemorrhage, and neurodegeneration as complications of DM [77].

Glucose metabolism for energy begins from glycolysis, during which glucose is converted to pyruvate, which can be further irreversibly transformed to acetyl-coenzyme A. Acetyl-coenzyme A may further react with oxalacetate to begin the tricarboxylic acid cycle (TCA)—an eight step enzymatic process coupled with nicotinamide-adenine dinucleotide (NAD) conversion to the reduced form (NADH), and flavin adenine dinucleotide (FAD) conversion to the reduced FADH. NADH and FADH transfer free electrons to the respiratory chain, which is coupled with ATP synthesis. Oxidative glycolysis of one glucose molecule, with H_2_O and CO_2_ as final products, can yield six NADH molecules, two FADH molecules and two ATP molecules, which is the equivalent of thirty-six ATP molecules. Aerobic glycolysis also provides the production of reducing equivalents, such as NADH and NADPH, which counteract oxidative stress caused by endogenous- and exogenous-reactive oxygen species (ROS) [78,79,80]. In normal conditions, the bioenergetic demand of mechanisms transporting various substances across the BBB is provided directly or indirectly by ATP. Neurons and astrocytes cooperate to fulfill their energy requirements. Glycolysis occurs mainly in astrocytes, while TCA: in neurons. Each type of cell contains both sets of enzymes, and metabolites produced in the course of glycolysis may enter other metabolic pathways, such as the pentose phosphate pathway (PPP), hexosamine biosynthetic pathway (HBP), protein kinase C pathway (PKC) and AGE biosynthesis pathway. Glucose may also enter the polyol biosynthetic pathway, being finally isomerized to fructose [78,79,80].

## 3. Correlations between Hyperglycemia and Oxidative Stress within BBB

Hyperglycemia can result in the excessive influx of glucose to the cells, which especially refers to insulin-independent cells, and may result in an increased ROS production through accelerating TCA reactions with the inhibition of the respiratory chain and ATP synthesis at the same time. This “hyperglycemic stress” is due to the fact that TCA reactions are coupled with transforming NAD into NADH, and in normal conditions, this NADH is used to transfer free electrons to the respiratory chain. In hyperglycemia, however, intracellular ATP level is already high, which inhibits the rate of respiratory chain reactions and ATP synthesis. Thus, there are too many NADH molecules in the mitochondria, and those NADH molecules may finally transfer the electrons into atomic oxygen, producing ROS. An increased ROS production in endothelial cells, which are susceptible to this kind of stress, results in microangiopathies and macroangiopathies as complications of DM [78,79,81,82]. In addition, this correlation between hyperglycemia and an increased ROS production may be essential for elucidating the pathogenesis of neurodegeneration as a possible complication of DM [83,84].

On the other hand, both acute and chronic hyperglycemia may stimulate the activity of PPP in cultured astrocytes, which is a source of reduced glutathione (GSH)—a co-enzyme for glutathione peroxidase, inactivating ROS. An increased activity of PPP promotes a reduced glutathione regeneration, thanks to which the cells are capable of counteracting oxidative stress [85]. In accordance with this fact, the nuclear factor erythroid 2-related factor 2 (Nrf2) transcription factor has been found to translocate to the cell nucleus, together with the binding immunoglobulin protein (BiP). Nrf2, taking part in cell defense against oxidative stress, is usually found in the cytoplasm, but is translocated to the cell nucleus in response to oxidative stress, thus initiating an antioxidative response dependent on the stimulation of expression of some endogenous antioxidant enzymes, such as NADPH-dependent quinone oxidoreductase, heme oxygenase, and glutathione S-transferase [85].

An increased ROS production in the course of hyperglycemia inhibits GADPH activity, and thus promotes glucose entering alternative metabolic pathways (glyceraldehyde 3-phosphate processing to protein kinase C (PKC) activating metabolites and the AGE-producing metabolic pathway) [76,85] (Figure 2).

Both endothelial cells and astrocytes use insulin-independent facilitative glucose transporter 1 (GLUT1). These cells, overloaded with glucose in hyperglycemic conditions, show mitochondrial dysfunction in which more than the usual electrons are directly transferred to O2 to generate reactive oxygen species (ROS) in the electron transport chain. Thus, hyperglycemia-driven mitochondrial tricarboxylic acid (TCA) cycle and its intermediates orchestrating mitochondrial oxidative phosphorylation are a significant center for ROS production. An increased ROS production, in turn, causes the inhibition of glyceraldehyde 3 phosphate dehydrogenase (GADPH) activity, and thus promotes glucose entering alternative metabolic pathways: glyceraldehyde 3-phosphate processing to protein kinase C (PKC) activating metabolites, and the advanced glycation end products (AGE)-producing metabolic pathway. Other glucose metabolic pathways are also overloaded, including the polyol pathway, pentose phosphate pathway (PPP), and hexosamine biosynthetic pathway (HBP). The last of the mentioned metabolic pathways is used for synthesis uridine diphosphate N-acetylglucosamine (UDP-GlcNAc), a nucleotide sugar and a coenzyme in metabolism.

* interdependent diabetic complications at the level of the brain-blood barrier (BBB), including micro- and micro-angiopathies, as well as neurodegenerative disorders.

Angiogenic edema occurring in the course of strokes, accompanied by hyperglycemia, has been found to result mainly from the excessive activation of the β isoform of PKC (PKCβ). Further activation of PKC promotes increased BBB permeability through ZO-1 phosphorylation, an impaired function of tight junctions, and a raised expression of VEGF [91]. Increased intracellular AGE concentrations may damage the cells through AGE-dependent modification of various proteins, affecting their interactions with the surface components of cell membranes (e.g., integrins) and receptors for advanced glycation end products (RAGE). It can refer to macrophages, endothelial cells, and smooth muscle cells. RAGE activation enhances ROS production, which in turn activates NF-κB dependent metabolic pathways, promoting the expression of pro-inflammatory mediators [78,79,92,93] and enhancing the innate immunity-dependent inflammatory response. As a matter of fact, it has been found that in people suffering from AD, AGE accumulation promotes neuronal death and degeneration, which confirms a hypothesis that DM can increase the risk of AD and dysglycemia, and can be detrimental to the BBB. In addition, oxidative stress activates matrix metalloproteinases, such as MMP-1, MMP-2 and MMP-9, which is accompanied by the reduced activity of their tissue inhibitors (TIMP-1 and TIMP-2), and occurs in a tyrosine kinase-dependent manner [83].

Although all insulin independent cells are exposed to an increased glucose concentration in the course of DM, only some kinds of cells become damaged in a hyperglycemia-dependent manner (e.g., retinal cells, endothelial cells), probably because they cannot reduce the expression of glucose transporters. As it has been mentioned above, hyperglycemic cell damage is dependent on five mechanisms: ❶—an increased influx of metabolites in the polyol pathway; ❷—an increased intracellular production of AGE; ❸—an increased expression of RAGE; ❹—activation of protein kinase C; and ❺—an increased influx of metabolites in the hexosamine biosynthetic pathway. All these mechanisms share the same downstream effector, which is increased by mitochondrial ROS production [78,94].

## 4. Role of HMGB1 as RAGE Ligand in Detrimental Effects of DM towards the CNS

The high-mobility group box 1 (HMGB1) protein is a non-histone chromosomal protein [95] regulating gene transcription through binding to DNA or chromatin, mediated by specific receptors, including RAGE and TLRs [96,97,98,99]. Because of its binding to RAGE and TLR4, HMGB1 may act as a pro-inflammatory mediator taking part in the pathogenesis of neurodegenerative diseases, such as AD [100,101]. RAGE, initially discovered as binding advanced glycation end products, can also bind other ligands, including beta-amyloid (Aβ) peptide, S100 proteins, and HMGB1 [102,103,104]. Some research studies suggest that hyperglycemia and insulin resistance underlying type 2 DM increase HMGB1 and RAGE expression both in diabetic mice and humans [105,106,107]. Furthermore, it has been found that, in the course of DM complications, HMGB1 activates NF-κB signaling pathways through interactions with RAGE and TLR4 [96] (Figure 3).

It was demonstrated that hyperglycemia and insulin resistance may increase expression of both, high-mobility group box 1 protein (HMGB1) and receptors, for advanced glycation end products (RAGE). The HMGB1 protein is a non-histone chromosomal protein regulating gene transcription through binding to DNA or chromatin, mediated by specific receptors, including RAGE and Toll-like receptors (TLRs). Therefore, the activation of HMGB1-RAGE-TLR4 axis in type 2 DM may induce an inflammatory response via the NF-κB signaling pathway. Pro-inflammatory cytokines, activated-matrix metalloproteinases, and reactive oxygen species (ROS) accumulating at the border of the brain and the capillary (vascular) compartments, are responsible for the degradation of tight junction proteins, such as zonula occludens-1 (ZO-1) and claudin-5. Consequently, cerebral micro-vessel leakage leads to increased BBB permeability with glial activation and the infiltration of immune cells into the brain parenchyma. Cognitive and memory impairments caused by chronic inflammation and oxidative damage are responsible for the clinical picture of diabetic encephalopathy. AGE—advanced glycation end products; Aβ—beta-amyloid; MyD88—myeloid differentiation primary response 88 (adapter protein); JNK—c-Jun N-terminal kinase; p50/p65—NF-κB heterodimer.

Recent studies suggest that, both in the brains of AD patients and in the CSF samples collected from them, HMGB1 concentration is elevated, just like in a mouse model of AD [109,110]. Several in vitro studies have revealed that the stimulation of HMGB1-RAGE-TLR4 signaling pathway promotes hippocampal neuron damage and memory loss in the course of AD [111,112,113]. In addition, RAGE interactions with HMGB1 are correlated with axonal overgrowth and neuroinflammation [104,114,115]. Another study has found that HMGB1 and TLR4 interaction promotes hippocampal neuron death in patients with DM [116]. HMGB1 activates astrocytes and promotes the release of proinflammatory cytokines, as well as stimulates inducible nitric oxide synthase (iNOS) expression in cortical astrocytes, thus stimulating TLR4 signaling [117]. Interaction between HMGB1, RAGE, and TLR4 promotes Aβ aggregate accumulation, stimulates neuroinflammation, dampens insulin-dependent signaling and impairs spatial memory [118,119]. Interactions between HMGB1, RAGE, and TLR4 are related to both DM and AD associated complications, such as Aβ accumulation, neuroinflammation, insulin-dependent signaling, memory deficits, and microglial cells activation.

In the course of AD, there is an increased activity of metalloproteinases, which can disrupt BBB integrity, thus contributing to neuronal and cognitive dysfunctions [120,121]. DM and dementia share some common features, such as severe and chronic neuroinflammation, brain-insulin resistance, an overaccumulation of Aβ and a disrupted BBB integrity [122].

Elevated levels of HMGB1 are correlated both with type 2 DM and with hyperglycemia [123,124,125]. In addition, HMGB1 impairs axonal growth through its interaction with RAGE [104,126], which may impair cognitive functions [127]. However, HMGB1 promotes Aβ accumulation and disrupts BBB integrity [128,129]. On a mouse model of AD, HMGB1 promotes axonal degeneration through myristoylated alanine-rich protein kinase C substrate (MARCKS) protein phosphorylation, which is dependent on TLR4 signaling [128]. TLR4, a transmembrane protein belonging to the pattern-recognition receptors (PRR) family, often takes part in innate immunity-dependent inflammation which has been correlated with AD-associated pathology. NF-κB signaling, which is downstream to TLR4 activation, promotes biosynthesis and the release of pro-inflammatory cytokines [130]. TLR4 expression is markedly increased in the brains of AD patients, which promotes amyloid-peptide binding and phagocytosis by microglial cells [131,132]. In the course of Aβ-induced neuroinflammation, HMGB1 can be localized in hippocampal neurons, where it is co-responsible for AD progression through activating RAGE- and TLR4-dependent signaling pathways [101]. In AD patients, HMGB1 accumulates both in the extracellular space and intracellular space of some brain regions [133].

Studies on in vitro models of AD show that HMGB1 is activated upon Aβ injection, which is accompanied by pro-inflammatory cytokine release and NLRP3 inflammasome assembly in microglial cells [134]. Furthermore, extracellular HMGB1 can impair microglia-dependent Aβ clearance, thus promoting AD through interactions with RAGE and TLR4 [135,136]. The same interactions may take part in the impairment of memory formation in mice [137].

HMGB1, as RAGE ligand, may promote insulin resistance of the brain through activating the TLR4-JNK signaling pathway [119,138,139], as well as through stimulating the TNF-α dependent signaling pathway [140,141] (Figure 4).

Downstream signaling then activates the inducible transcription factors: NFAT5, AP-1, and NF-κB. A pro-inflammatory environment develops because the target genes for all these transcription factors include cytokines, such as IL-6, IL-1β, IL-18, and TNF-α. TNF-α, acting on its own receptor TNFR1, may interfere with insulin signaling through phosphorylation of some serine/threonine residues in IRS, especially IRS-1. Such an inhibitory phosphorylation of IRS-1 was also demonstrated for the components of the signaling pathways (marked in the figure with “*”). Interestingly, the activation of the NF-κB pathway could, in turn, induce the expression of HMGB1 and its receptors, forming a positive feedback loop to sustain inflammatory conditions. AC—astrocyte; AP-1—activator protein 1; BBB—blood–brain barrier; EC—endothelial cell; ERK1/2—extracellular signal-regulated kinases 1 and 2, also known as classical mitogen-activated protein (MAP) kinases; HMGB1—high mobility group box 1; IκB—inhibitory κB protein; IKKα, IKKβ, IKKγ—the members of the inhibitor of κB (IκB) kinase (IKK) family; INS—insulin; INSR—insulin receptor; ISF—interstitial fluid; p65/p50—nuclear factor kappa-light-chain-enhancer of activated B cells (NF-κB) heterodimer; JNK—c-Jun N-terminal kinase; MyD88—myeloid differentiation primary response 88 (adapter protein); NFAT5—nuclear factor of activated T cells 5; P—phosphorylation; Raf—rapidly accelerated fibrosarcoma; RAGE –– receptors for advanced glycation end-products; ROS—reactive oxygen species; Ser—serine; TLR4—Toll-like receptor 4; TNF-α—tumor necrosis factor alpha; ZO-1—zonula occludens-1 (also known as tight junction protein-1).

## 5. Summary and Conclusions

Insulin dependent signaling in the CNS seems to be crucial for the maintenance of cognitive functions, through the regulation of neurotransmitter release, synaptic transmission, and glucose uptake by neurons [144,145,146]. A disrupted brain–glucose metabolism, in combination with the insulin resistance of the hippocampus, may contribute to synaptic dysfunction, cognitive-function impairment, and the development of AD [147,148]. In addition, brain-insulin resistance reduces cerebral blood flow [149] and cerebral cortex perfusion, which leads to cognitive deficits [150]. Moreover, insulin resistance promotes excessive Aβ aggregate accumulation and the hyperphosphorylation of tau proteins, which results in cognitive impairment in patients with Alzheimer’s disease [148,151]. Normal activity of insulin-dependent signaling pathways facilitates Aβ aggregate clearance and inhibits senile-plaque formation [152]. Insulin resistance has been found to accelerate Aβ formation in the vicinity of presynaptic neuronal cell membranes [153,154] and has been correlated with activating the JNK-dependent signaling pathway with subsequent inhibitory phosphorylation of the insulin receptor substrate 1 (IRS-1) at S616 [155]. Thus, TLR4 activation by AGE or HMGB1 can increase the risk of dementia trough promoting brain-insulin resistance.

RAGE activation, by AGE or HMGB1, may exert a pro-inflammatory effect because it activates intracellular-signaling pathways stimulating NF-κB activity [101,156], while AGE themselves can induce a pro-inflammatory cytokine release, thus activating the innate immunity-dependent inflammatory response [157].

The sum of detrimental effects of dysglycemia towards the BBB consists of its pro-oxidative effect promoting ROS production [78,79], which in turn inhibits glyceraldehyde 3 phosphate dehydrogenase (GADPH) activity [78], thus redirecting glucose metabolism to AGE [78,94]. Subsequently, the excess of AGE may stimulate RAGE as well as some TLR receptors (e.g., TLR4), which induces NF-κB activity and promotes neuroinflammation [78,79,92,93]. Of note, pro-oxidative effects of dysglycemia can be neutralized with Nrf2 transcription-factor induction [158].

Detrimental effects of dysglycemia towards the BBB include: disrupting its integrity and increasing permeability, mainly through promoting oxidative stress [78,91]; inducing inflammatory response through RAGE and TLR4 activation; the redistribution of glucose transporters, such as GLUT-1; and related alteration of BBB permeability for choline and DHA [10,51]. These effects may, in turn, impair acetylcholine biosynthesis and the antioxidative defense that is dependent on vitamin C.

As a result of the effects mentioned above, DM can increase the risk of neurodegeneration and dementia. Firstly, through disrupting BBB integrity. Secondly, through promoting brain-insulin resistance with its detrimental effects on cognitive functions [122], Thirdly, through inducing the excessive production of some substances, such as AGE and HMGB1, which may promote neuroinflammation, thus abrogating the function of microglia and Aβ clearance [159,160]. Fourthly, through the increased production of AGE and HMGB1, which can directly stimulate Aβ aggregate production [128,135,161], thus accelerating their accumulation in the CNS. All of these effects may increase the risk of not only Alzheimer’s disease, but also vascular dementia and mixed-type dementia, the latter two being related to detrimental effects of prolonged hyperglycemia on endothelium, thus promoting cerebrovascular microangiopathy [162,163].

## Figures and Tables

**Figure 1 ijms-24-10069-f001:**
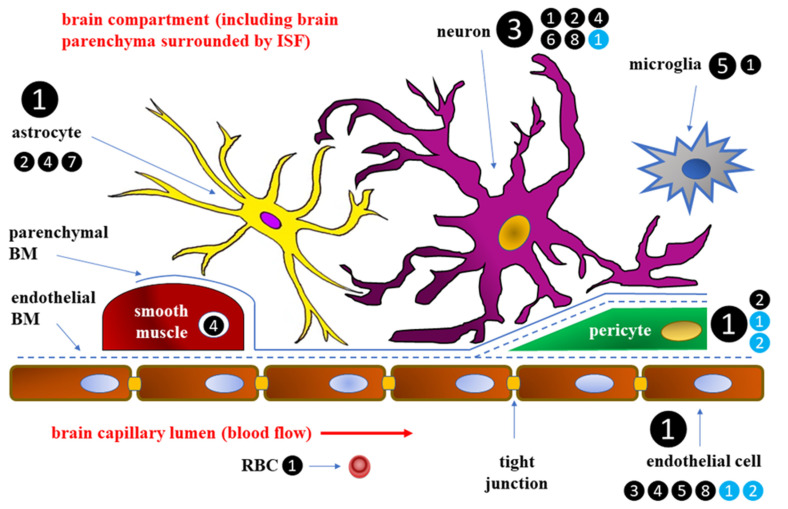
Schematic diagram of the blood–brain barrier (BBB) [8,9,11,12,17,27,28]. The numbers in black circles (1–8) correspond to the glucose transporters (GLUTs) 1–8, respectively; the numbers in blue circles (1–2) correspond to the sodium-glucose cotransporters (SGLTs) 1–2, respectively; the main glucose transporter for a given cell type under physiological conditions is marked with an enlarged symbol. BM—basement membrane; ISF—interstitial fluid; RBC—red blood cell.

**Figure 2 ijms-24-10069-f002:**
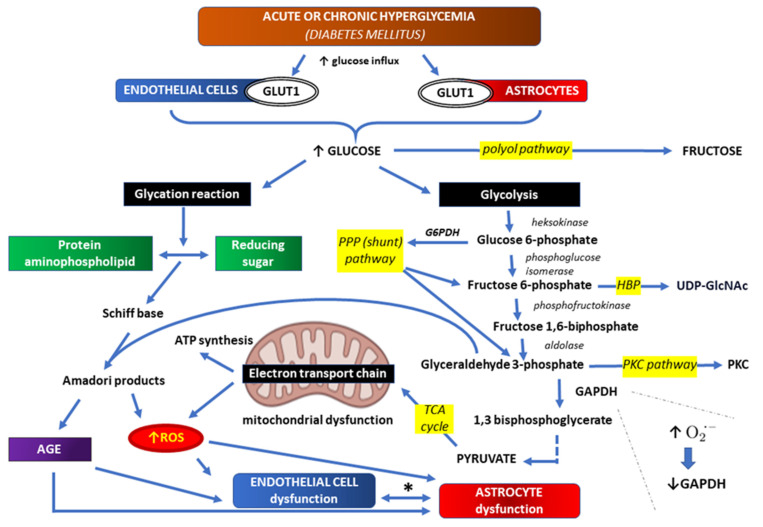
Hyperglycemia and oxidative stress in endothelial cells and astrocytes resulting from increased glucose influx [85,86,87,88,89,90].

**Figure 3 ijms-24-10069-f003:**
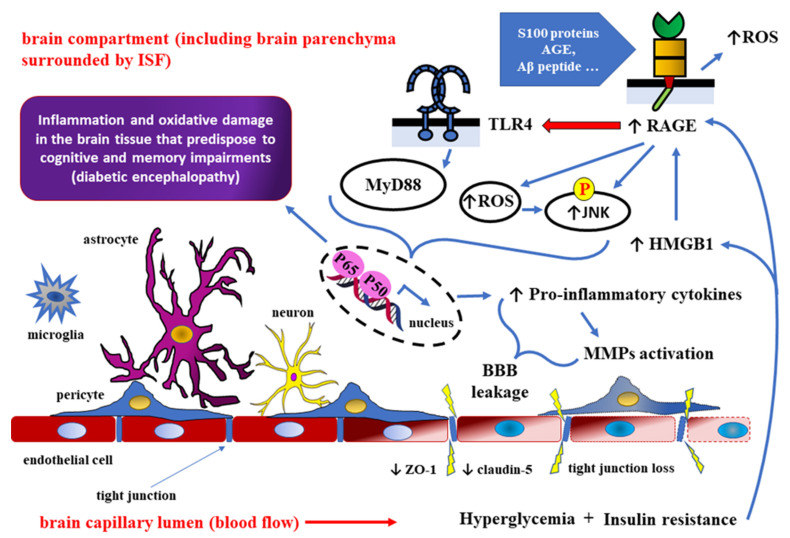
HMGB1-RAGE-TLR4 signaling in type 2 diabetes mellitus (DM) may compromise the integrity of the BBB and increase the risk of neurodegenerative diseases [95,96,97,98,99,100,101,102,103,104,105,106,107,108].

**Figure 4 ijms-24-10069-f004:**
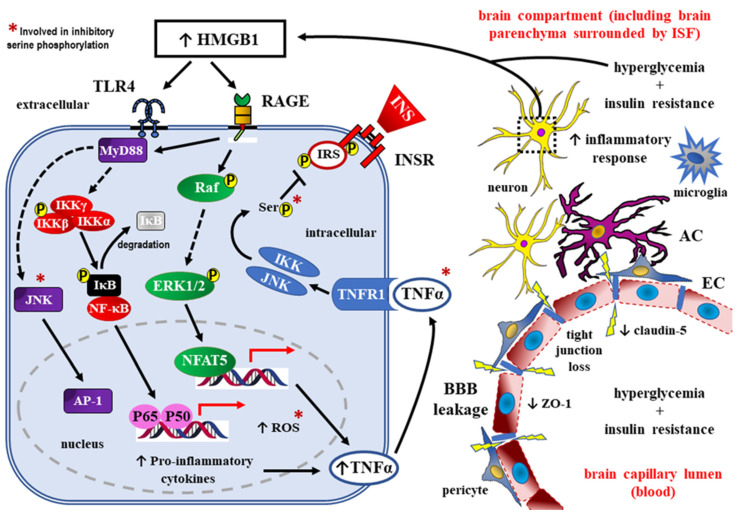
Increased HMGB1 expression in the brain micro-vessels of type 2 DM patients and, ultimately, in the brain compartment, may promote insulin resistance of the brain via activation of both RAGE and TLR4 signaling pathways [107,119,138,139,140,141,142,143].

## Data Availability

Not applicable. This review is based on already published data listed in the references.

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
