# Peer review of "Effects of Diabetes Mellitus-Related Dysglycemia on the Functions of Blood–Brain Barrier and the Risk of Dementia"

_ijms, 2023, doi:10.3390/ijms241210069_

Round 1

Reviewer 1 Report

Review report

Journal: IJMS (ISSN 1422-0067)

Manuscript ID: ijms-2446983

Type: Review

Title: Effects of diabetes mellitus related dysglycemia on the functions of blood-brain barrier and the risk of dementia

Authors: Mateusz WÄ…troba , Anna D. Grabowska , Dariusz Szukiewicz *

Section: Molecular Pathology, Diagnostics, and Therapeutics

Special Issue: Astrocyte-Endothelial Interactions at the Blood-Brain Barrier

General

The authors of this review article focus on the deleterious impact of dysglycemia, particularly hyperglycemia, a condition associated with diabetes mellitus, on the structure and functionality of the blood-brain barrier (BBB). The manuscript is well-written, easy to read, and logically organized.

The authors present an in-depth analysis of the potential consequences of long-term hyperglycemia on the central nervous system (CNS), linking this to an increased risk of neurodegenerative diseases and dementia. The manuscript proposes that hyperglycemia can lead to oxidative stress and inflammatory responses within the CNS, damaging cells and promoting neurodegeneration. Further, the authors discuss how advanced glycation end products (AGE) could exacerbate this damage via inflammatory pathways. Additionally, they highlight the potential for hyperglycemia to foster brain insulin resistance, contributing to the accumulation of Aβ aggregates and tau hyperphosphorylation, both well-recognized hallmarks of neurodegenerative diseases. In the broader context, the authors aim to shed light on the underlying mechanisms linking diabetes mellitus and BBB dysfunction to the pathogenesis of long-term complications in the CNS.

Minor points

The manuscript is of excellent quality, offering an in-depth, comprehensive analysis of the intricate interplay between Diabetes Mellitus (DM) and its disruptive influence on the Blood-Brain Barrier (BBB) and its associated implications for dementia risk. Thus, I advocate for the acceptance of this manuscript for publication.

However, I suggest a few enhancements that would further strengthen the manuscript.

1. Certain terms and concepts may need further explanation to increase accessibility to readers from various backgrounds. For example, a more detailed discussion on the blood-brain barrier (BBB), its structure (why is important in Diabetes?), and its roles could be beneficial. The same applies to dysglycemia and the specific physiological and biochemical changes it triggers.

2. While the review thoroughly discusses the existing literature, it might benefit from a more explicit section discussing ongoing research and future perspectives. This could include current gaps in knowledge, unresolved questions, or potential future therapeutic interventions based on the discussed mechanisms.

3. Given the global prevalence of diabetes and dementia, it may be worth discussing international perspectives, policies, or demographic data, to underscore the universality and urgency of the issue.

4. While the authors conclude the review mentioning Alzheimer's disease, including more information on other types of dementia could provide a more comprehensive understanding of the subject.

In conclusion, the manuscript offers a well-structured and compelling synthesis of the current understanding of the relationship between dysglycemia, BBB, and dementia risk. It is commendable for its clarity and depth, providing a robust foundation for further investigation into this crucial medical issue. Given the global burden of DM and dementia, this timely review is sure to be of significant interest to the readership.

Therefore, once the recommended improvements have been addressed, I support the acceptance of this manuscript for publication.

Reviewer 2 Report

Well written, synthetic paper on an original subject (glucose & brain).

Some questions :

- a glucose "memory" in the brain ?

- The effect of hyper (down regulation of GLUT) and increase in hypo are well described. Nowadays, thanks to CGM, we speak of glucose variability. What is the effect of glucose variations in the brain ? 

One concern line 249 : HIF 1 and VEGF do act synergistically 

Clear paper . Intersting subject.
